# Effects of Hydrogen Sulfide on Sugar, Organic Acid, Carotenoid, and Polyphenol Level in Tomato Fruit

**DOI:** 10.3390/plants12040719

**Published:** 2023-02-06

**Authors:** Yanqin Zhang, Fahong Yun, Xiaoling Man, Dengjing Huang, Weibiao Liao

**Affiliations:** College of Horticulture, Gansu Agricultural University, 1 Yingmen Village, Anning District, Lanzhou 730070, China

**Keywords:** flavor, soluble sugar, organic acid, phenolic acids, carotene, postharvest

## Abstract

Hydrogen sulfide (H_2_S) is known to have a positive effect on the postharvest storage of vegetables and fruits, but limited results are available on its influence in fruit flavor quality. Here, we presented the effect of H_2_S on the flavor quality of tomato fruit during postharvest. H_2_S decreased the content of fructose, glucose, carotene and lycopene but increased that of soluble protein, organic acid, malic acid and citric acid. These differences were directly associated with the expression of their metabolism-related genes. Moreover, H_2_S treatment raised the contents of total phenolics, total flavonoids and most phenolic compounds, and up-regulated the expression level of their metabolism-related genes *(PAL5, 4CL, CHS1, CHS2, F3H* and *FLS)*. However, the effects of the H_2_S scavenger hypotaurine on the above flavor quality parameters were opposite to that of H_2_S, thus confirming the role of H_2_S in tomato flavor quality. Thus, these results provide insight into the significant roles of H_2_S in tomato fruit quality regulation and implicate the potential application of H_2_S in reducing the flavor loss of tomato fruit during postharvest.

## 1. Introduction

Hydrogen sulfide (H_2_S), a colorless acid gas with an odor of rotten eggs, can freely pass through biofilms and participate in signal transduction as a signal molecule [1,2]. Presently, it is considered the third gaseous signaling molecule, followed by nitric (NO) and carbon monoxide (CO) [3,4]. It is well known that the production of endogenous H_2_S in plants is mainly through L-cysteine desulfhydrase (L-DES), D-cysteine desulfhydrase (D-DES), sulfite reduction enzymes (sulfite reductase SiR), cyanoalanine synthase (CAS), and cysteine synthase (cysteine synthase, CS) metabolism pathway [5,6,7]. In recent years, many studies have found that H_2_S participates in many physiological processes in plants, e.g., seed germination, leaf and root organ development, stomatal movement, photosynthesis, material metabolism and postharvest preservation [4,8].

For example, the germination time of wheat, corn and soybean seeds treated with exogenous H_2_S were significantly earlier than that of natural germination seeds [9]. H_2_S also plays important roles in the process of delaying senescence postharvest and regulating the quality of horticultural products, e.g., fruits and vegetables [10,11,12].

Recently, numerous studies have begun to focus on the preservation effect of H_2_S, which may be caused by the fact that a low-concentration of H_2_S has been proven to be harmless to humans, and it also does not easily remain in the body [12,13,14]. H_2_S significantly inhibits the respiration of fruits, thereby reducing the consumption of nutrients and maintaining good nutritional quality [13,15]. Exogenous H_2_S significantly increases the activity of antioxidant enzymes in strawberries, which scavenges ROS and delays senescence [13]. H_2_S also inhibits the activity of polygalacturonase and relieves the degradation of cell wall polygalacturonic acids, thereby delaying the softening of strawberries and prolonging their shelf life [16]. H_2_S alleviates the ripening and senescence of kiwifruit and banana by antagonizing the effect of ethylene through the inhibition of the ethylene synthesis pathway [17,18]. Additionally, H_2_S could maintain the external color of agricultural crops (i.e., strawberry, mulberry, Lanzhou lily-bulb scales, broccoli florets and water spinach) and preserved the color of fresh-cut lotus root, potato, yam, lettuce and pumpkin and so on [4,19,20]. In a word, H_2_S could delay the ripening and senescence of agricultural products, e.g., fruits and vegetables.

Tomato (*Solanum lycopersicum*) is one of the most popular vegetables in the world and has the second most consumed among vegetables [21]. Tomatoes can enhance human immunity to prevent the occurrence of various diseases, such as cancer and cardiovascular disease [22,23]. The nutritional quality of tomatoes and tomato variety, ripening period and storage conditions has been widely studied [24,25]. The nutritional importance of tomatoes is largely explained by their various health-promoting compounds, including vitamins, carotenoids, and phenolic compounds [26,27]. Tomatoes are rich in antioxidants; vitamins such as Vitamins A, B, C and E; and active substances such as lycopene and phenolic compounds that act in the scavenging of free radicals, slow down atherosclerosis and prevent heart disease [28,29]. Tomatoes are also rich in carotenoids, representing the main source of lycopene in the human diet [30]. Carotenoids and polyphenolic compounds contribute to the nutritional value of tomatoes and improve their functional attributes and sensory qualities, including taste, aroma, and texture [28,31]. Moreover, the fruit of tomatoes also contains large amount of sugar, protein, cellulose, pectin, organic acids, fat, and mineral (copper, iodine, zinc, etc.) [32]. Despite its high nutritional value and high global production (about 5,051,983 ha cultivated area with an estimated production of 186.821 MMT in 2022) [33], postharvest losses as high as 25–42% make its production unprofitable [34]. Ripeness, rotting and disease causes flavor loss in tomatoes [35]. The postharvest quality of tomatoes is determined by a variety of preharvest factors such as tomato varieties, cultivation conditions, and management techniques [36]. Some postharvest factors, including storage temperature and relative humidity, physical handling, and combination gases, have been identified [36,37].

Therefore, improving the quality of tomatoes has become an urgent and common desire of producers and consumers.

H_2_S fumigation is gradually used in the preservation of agricultural products such as grape, banana, peach, apple, kiwifruit, broccoli and tomato [7,38,39]. Previous studies on preservation with H_2_S mainly focus on two aspects: antagonizing ethylene and enhancing antioxidant [18,40]. Among them, starch, sugar, ascorbic acid, total phenol and flavonoids are the most widely studied [19,41]. However, no comprehensive analysis has been reported on the regulatory roles of H_2_S in the flavor quality in tomato fruit. In this study, therefore, tomatoes (*S*. *lycopersicum* ‘Ailsa Craig’) were used as experimental materials to study the effects of H_2_S on postharvest flavor and nutritional quality, especially the classification and content of polyphenols, thereby providing insight into the significant roles of H_2_S in tomato fruit quality regulation and implicating the potential application of H_2_S in maintaining tomato fruit nutrient.

## 2. Results

### 2.1. Effects of H_2_S on Soluble Sugar, Soluble Protein and Organic Acids Content in Tomato Fruit

Compared with the control (H_2_O treatment), the content of soluble sugar, fructose and glucose in H_2_S treatment was significantly decreased, whereas it was significantly increased with HT treatment (Figure 1a). In contrast, soluble protein content was higher with H_2_S treatment but lower with HT treatment (Figure 1b). H_2_S treatment significantly increased the content of organic acids, malic acid and citric acid compared to the control (Figure 1c). However, compared to the control, HT treatment significantly decreased organic acids, malic acid and citric acid content.

### 2.2. Effects of H_2_S on the Expression of Genes Related to Soluble Sugar and Organic Acids Metabolism in Tomato Fruit

As the glucose and fructose metabolism-related genes *pyrophosphate--fructose 6-phosphate 1-phosphotransferase subunit beta-like* (*PFPP*), *ATP-dependent 6-phosphofructokinase 6* (*ATPPF*), *fructose-bisphosphate aldolase* (*FBA6*), *glucose-6-phosphate isomerase* (*GPI*), *glucose-6-phosphate 1-dehydrogenase, cytoplasmic isoform* (*GPD1*) and *probable 6-phosphogluconolactonase 2* (*GPD2*) could affect glucose and fructose content in tomato fruit, their transcriptional levels were determined. Compared to the control, H_2_S treatment significantly decreased the expression level of fructose metabolism-related genes *PFPP*, *ATPPF* and *FBA6*, and glucose metabolism-related genes *GPI*, *GPD1* and *GPD2* (Figure 2a,b). Compared to the control, HT treatment induced higher expression of *PFPP*, *ATPPF*, *FBA6*, *GPI*, *GPD1* and *GPD2* genes (Figure 2a,b). The transcription level of malic acid and citric acid metabolism-related genes *containing malate dehydrogenase* (*MDH*), *NADP-malic enzyme* (*ME1*), *cytosolic NADP-malic enzyme* (*ME2*), *citrate synthase, glyoxysomal* (*CSG*), *citrate synthase 3, peroxisomal-like* (*CS3*) and *citrate synthase, mitochondrial* (*CSM*) were significantly up-regulated by H_2_S and were obviously down-regulated by HT treatment when comparing to the control (Figure 2c,d).

### 2.3. Effects of H_2_S on Ascorbic Acid, Total Phenols, Total Flavonoid and Carotenes Content in Tomato Fruit

A higher content of ascorbic acid, total phenols and total flavonoids was observed in the tomato fruit treated with exogenous H_2_S. However, less of them accumulated with HT treatment than with the control (Figure 3a). The content of carotene, lycopene and lutein was significantly reduced by H_2_S treatment in comparison with the control, while HT treatment significantly increased their content when compared to the control (Figure 3b).

### 2.4. Effects of H_2_S on the Expression of Genes Related to Polyphenols, Flavonoid, Ascorbic Acid and Carotene Metabolism in Tomato Fruit

The expression level of *phenylalanine ammonia-lyase 5* (*PAL5*), *4-coumarate-CoA ligase* (*4CL*), *monodehydroascorbate reductase* (*MDHAR*) and *dehydroascorbate reductase 1* (*DHAR1*) genes was significantly higher with H_2_S treatment than with the control. However, the expression level of these genes with HT treatment was inhibited (Figure 4a,c). Similarly, the flavonoid metabolism-related genes *chalcone synthase* (*CHS1*), *chalcone synthase* (*CHS2*), *flavanone 3-dioxygenase* (*F3H*) and *flavonol synthase* (*FLS*) were up-regulated with H_2_S treatment but down-regulated with HT treatment (Figure 4e). The transcription level of carotene and lycopene metabolism-related genes including *zeta-carotene desaturase* (*ZDS*), *zeta-carotene desaturase* (*CCD7*), *lycopene epsilon-cyclase* (*CrtL-e-1*), *phytoene synthase 1, chloroplastic* (*psy1*), *lycopene beta-cyclase* (*LCY1*) and *phytoene desaturase* (*PDS*) was reduced by H_2_S treatment (Figure 4b,d). Compared to the control, the expression of carotene and lycopene metabolism-related genes was higher in HT-treated fruit (Figure 4b,d).

### 2.5. Effects of H_2_S on Polyphenol Content in Tomato Fruit

Polyphenolic compounds, including protocatechuic acid, P-hydroxybenzoic, chlorogenic acid, gallic acid, 4-coumaric acid, ferulic acid, benzoic acid, cinnamic acid, gentianic acid, caffeic acid, erucic acid, cynarin, kaempferol, rutin and quercetin, were identified (Table 1). In the control, the top ten phenolic compounds were rutin (153.656 μg/mL), gentianic acid (34.519 μg/mL), quercetin (16.978 μg/mL), benzoic acid (8.683 μg/mL), gallic acid (8.484 μg/mL), kaempferol (7.887 μg/mL), chlorogenic acid (5.358 μg/mL), ferulic acid (5.332 μg/mL), caffeic acid (5.179 μg/mL) and cinnamic acid (5.144 μg/mL). Obviously, in comparison with the control, H_2_S treatment significantly raised the accumulation of most of the phenic acids, such as protocatechuic acid, chlorogenic acid, gallic acid, ferulic acid, benzoic acid, and gentianic acid. However, the content of most of the phenic acids, including protocatechuic acid, chlorogenic acid, gallic acid, ferulic acid, benzoic acid, and gentianic acid was lower in HT-treated tomatoes than in the control tomato. In contrast, H_2_S treatment significantly decreased the content of flavonoids such as kaempferol, rutin and quercetin, while the content of flavonoids such as kaempferol, rutin and quercetin was higher in HT-treated tomatoes.

### 2.6. Effect Sizes of H_2_S on Flavor Quality in Tomato Fruit

As shown in Figure 5, the results of the cluster analysis and the effect size suggests that the effect of H_2_S on tomato fruit quality was mainly divided into two categories. The first type is such that their effect value was less than zero in Ln(H_2_S/H_2_O), but greater than zero in Ln(HT/H_2_O); that is, H_2_S might play a negative regulatory role on contents such as fructose, glucose, quercetin, lutein, soluble sugar and rutin. The other category is that which is up-regulated by H_2_S. In this, the content of soluble protein, organic acids (including malic acid and citric acid), ascorbic acid, total flavonoid, and some phenolic compounds such as ferulic acid, benzoic acid, gallic acid, chlorogenic acid, erucic acid and cynarin were significantly up-regulated by H_2_S (Figure 5).

## 3. Discussion

Tomato, as one of the world’s favorite fruits, is an important source of energy, minerals, vitamins, flavonoids, phenolics and other phytochemicals in human diets [42,43]. However, the postharvest ripening of tomato fruit at room temperature is very rapid, resulting in a rapid decline in fruit firmness and nutrient loss, wherein it then enters the fruit senescence stage [44]. Based on this, approaches have been developed to delay the process of fruit ripening and lessen the nutrition loss, including refrigeration control, atmosphere modification (fumigation with exogenous NO, H_2_S, H_2_O_2_, etc.) and chemical substance treatment such as ethylene inhibitor, methyl jasmonate and salicylic acid [45,46,47,48,49]. In the present study, we applied H_2_S produced by NaHS to explore its function in tomato flavor and nutrient regulation after harvest. Sugar, as an energy substance and signaling molecule, affects fruit sweetness and ripening [50,51]. The content of sugar in tomato fruit was found to be regulated by the addition of H_2_S. In strawberries and fresh-cut pears, H_2_S maintained a higher level of sugar reduction, while H_2_S inhibited the increase of soluble sugar content in kiwifruit [13,39,52]. Here, we also found that exogenous H_2_S treatment declined the content of soluble sugar. The metabolism and accumulation of sugar are important factors for fruits organoleptic and nutritional quality [53,54]. During cultivated tomato fruit’s maturity stage, fructose and glucose (as the main sugars) were increased with the consumption of sucrose [55], which explains the lower level of fructose and glucose in H_2_S-treated tomato fruit compared with the water treated ones (Figure 1a). The changes in fructose and glucose metabolism-related gene expression under different treatments were in keeping with the changes in their contents, suggesting that H_2_S might regulate sugar levels by regulating the expression of sugar metabolism-related genes. Soluble protein content is one of the important indexes for evaluating fruit quality and nutrition. The ripening and senescence of a tomato is accompanied by the reduction of nutrition-related metabolites including soluble proteins [56]. Hu et al. [13] showed that the degradation of soluble proteins in strawberries was inhibited by the exposure of NaHS at the early stage of storage, and the soluble protein content in H_2_S treatment remained higher than that in the control during the entire storage period. Similarly, our results suggested that tomatoes under H_2_S treatment contained higher soluble protein content. Because acids also influence the sweetness of tomato fruit, the content of organic acid was measured in the study. Compared to the control, the organic acid content in the tomato fruit was boosted by H_2_S but decreased by HT. H_2_S has also raised the citric acid content in tomato [57] and kiwifruit [17]. In tomato, citric and malic are two primary acids, and citric acid is half as acidic as malic [58]. Thus, combining the expression of citric and malic metabolism-related genes, we found that the expression of both citric (*CSG, CS3* and *CSM*) and malic (*MDH, ME1* and *ME2*) metabolism-related genes was positively correlated with their accumulation and was increased by H_2_S while decreased by HT. With the exception of sugar and acids, ascorbic acid, flavonoids, carotenoids, and phenols are also important nutrients in tomato fruit. In broccoli and strawberry, H_2_S maintained higher ascorbic acid (or vitamin C) [13,38]. This was also found here in the tomato fruit (Figure 3a). Tomato flavor and nutrient are produced by a combination of acids, sugar, and volatiles [59]. Besides its functions in pigments and nutrients, carotenoids also act as important precursors of volatile flavor compounds in plant [60]. Yao et al. [41] reported that the carotenoid content in tomato fruit at the white mature stage was higher in C_2_H_4_-H_2_S co-treatment than that in C_2_H_4_ treatment, suggesting that H_2_S could maintain better nutritional quality than C_2_H_4_ treatment alone. Here, we found that H_2_S treatment deceased the carotenoid content in tomato. The similar result was reported in grape, kiwifruit and banana [18,61,62]. Meanwhile, H_2_S also inhibited the expression of carotenoid metabolism-related genes including *ZDS, CCD7, CrtL-e-1, psy1, LCY1* and *PDS*.

Phenolics and flavonoids are major secondary metabolites in fruits and vegetables and are associated with numerous health-promoting properties for their antioxidant ability [63,64]. Here, H_2_S enhanced total phenols content, while HT decreased the content. In contrast, in the study of H_2_S-alleviated broccoli postharvest senescence, total phenol content was found to be slackened by H_2_S fumigation [38]. This might be due to the difference of the species. In tomato fruit, the content of phenolic is influenced by the ripening stage; the phenolic accumulation in green and medium-ripened fruit was higher than that in fully ripened fruit [65]. Phenylalanine ammonialyase (PAL) is the first key enzyme in the phenylpropanoid pathway [64]. In *Arabidopsis*, *PAL1* and *PAL2* could increase the accumulation of flavonoids, anthocyanins, and sinapic acid [66]. C4H and 4CL are another two important enzymes in the phenylpropanoid pathway [67]. In the present study, exogenous H_2_S promoted the expressions of the *PAL, C4H* and *4CL* genes in the tomatoes, which was consistent with the effect of H_2_S on total phenols and flavonoid content. In grape pulp, H_2_S was reported to maintain a higher content of phenolics [61]. In addition, the expression level of *PAL*, *C4H* and *4CL* was inhibited by the addition of H_2_S scavenger HT. The content of flavonoid, as the major phenolic in tomato, was reported to be elevated by H_2_S in hawthorn fruit [68]. Moreover, H_2_S maintained a high content of phenolics and flavonoids in banana fruit [18]. H_2_S was reported to alleviate the decrease of nonenzymatic antioxidant components such as carotenoids, ascorbate and flavonoids in broccoli to prolong the freshness during postharvest storage [38]. Here, we found that both flavonoid and transcriptional level and their metabolism-related gene expression were increased by H_2_S treatment but were decreased by HT treatment compared to the control.

Currently, the identification of phenolic compounds in plants is still in progress, and tomatoes, as one of the sources of a variety of phenolic compounds, are also under the scope of extensive studies [64]. Here, the top ten phenolic compounds identified in tomatoes under the control were not completely consistent with those reported by Tao et al. [46] and Liu et al. [69], who suggested that caffeic acid, (+)-catechin, chlorogenic acid, isoquercitrin, syringic acid, p-coumaric acid, rutin and sinapic acid are the key phenolic compounds in tomato fruit. This difference in the top phenolic compounds may be account for by the cultivar, cultivation, handing and storage methods [70]. In addition, we first analyzed the influence of H_2_S and HT on the content of various phenolic compounds. Most of the phenic acids, such as protocatechuic acid, chlorogenic acid, gallic acid, ferulic acid, benzoic acid and gentianic acid, were raised/up-regulated significantly by H_2_S treatment, while their content was declined/down-regulated by HT. In contrast, the content of flavonoids such as kaempferol, rutin and quercetin was decreased/down-regulated by H_2_S treatment but was significantly increased/up-regulated by HT treatment.

## 4. Materials and Methods

### 4.1. Plant Material and Treatment Conditions

The cultivar Ailsa Craig tomatoes (*S. lycopersicum*) were collected from the Glasshouse of Horticulture, Gansu Agricultural University, Lanzhou, China. Tomatoes without mechanical damage and of equal size were selected and harvested at mature green (approximately 30 days after anthesis, brix (about 0.8%), color (green ripening) and weight (about 10 g for every tomato)) stage for the experiments. A sodium hydrosulfide (NaHS) aqueous solution of 0.90 mM as a H_2_S donor and hypotaurine (HT; 100 mM) as the scavenger of H_2_S were used in the study. The treatment method of tomato fruit was as follows: seven tomato fruits were randomly selected as a group and placed in a petri dishes (12 cm × 12 cm), and then two groups of tomatoes were placed in a sealed plastic box (volume 3 L) with perforated baffle. Each treatment was repeated three times, and there were 42 tomatoes in each repetition. The box was sealed quickly when the NaHS solution (0.90 mM) was added to the beaker, which was at the bottom of the plastic cling box. In accordance with Hu et al. [12], H_2_S gas released from NaHS solution (0.90 mM) reached 1.00 × 10^−10^ mol/L within 30 min and remained stable throughout the following 24 h. Then, the NaHS solution was replaced with distilled water after 24 h. The remaining two groups were injected with the same volume of H_2_O (the control) or 100 mM HT solution. Finally, tomatoes were stored at 25 °C with a relative humidity of around 75−85%, and the samples were collected on the 5th day after treatment (the sampling time was based on previous research results in our laboratory). The seeds in the tomato fruit were removed, and the flesh ones were frozen in liquid nitrogen immediately and stored in a −80 °C freezer.

### 4.2. Determination of Soluble Sugar and Soluble Protein Content

Soluble sugar was determined by using the anthrone colorimetry method [71]. Approximately 1.0 g of tomato fruit sample was ground in liquid nitrogen, and the homogenate was extracted in 10 mL of distilled water and in boiled water twice for 30 min each. Then, the homogenate was filtered into supernatant and fixed in a 25 mL volumetric flask for determination. The supernatant (0.5 mL) was mixed with distilled water (1.5 mL), anthrone-ethyl acetate reagent (0.5 mL) and concentrated sulfuric acid (5.0 mL), and incubated in boiled water for 1 min. Then, the reaction solution was measured for its absorbance at 630 nm with an ultraviolet spectrophotometer (UV-1800, Shimadzu, Kyoto, Japan). The soluble sugar content was then obtained using glucose as a standard and by comparing the standard curves (y = 0.0068x + 0.0008; R^2^ = 0.9996).

The soluble protein content was determined by the method of Coomassie Brilliant Blue G250 staining. About 1.0 g of tomato fruit was ground with distilled water and extracted at room temperature for 45 min. The homogenate was centrifuged at 4000× *g* for 10 min, and the supernatant was collected for measurement. The supernatant (0.1 mL), distilled water (0.9 mL) and Coomassie Brilliant G-250 (5 mL) were mixed and stood for 2 min. Then, the absorbance of supernatant was measured at 595 nm after complete reaction. Soluble protein content was measured using bovine serum albumin (BSA) as a standard via the specific reaction of coomassie brilliant blue G-250 dye with maximum absorbance at 595 nm (y = 0.0121x + 0.2945; R^2^ = 0.9984), and the result was expressed as mg/g.

### 4.3. Determination of Sugar and Acid Fraction

Before the determination of glucose, fructose, organic acid, citric acid and malic acid, the extracts were filtered through a 0.22 μm filter membrane. The sugar fraction was determined according to the method of Gomez et al. [72]. Chromatographic separation was performed in a NH_2_-Ms column, RI Detector (L-2490) with a differential detector at 35 °C, and the mobile phase was acetonitrile-water (80:20). The injection volume was 10 μL, and the analyses were performed at a flow rate of 1 mL/min.

The determination of the acid fraction was based on the method of Gao et al. [73]. The detector was a diode array detector, and the column was a C18 reversed-phase column at 30 °C. The mobile phase was 0.04 mM KH_2_PO_4_ (pH 2.4) solution, the injection volume was 10 μL, and the analyses were performed at a flow rate of 0.5 mL/min. Three biological replicates were performed for each of the above components, and the content of each component was calculated based on the sample peak area and the standard curve. The fructose content was then obtained using fructose as a standard and comparing the standard curves (y = 212,533x + 264,099; R^2^ = 0.9965), and the glucose content was then obtained using glucose as a standard and comparing the standard curves (y = 181,019x + 275,857; R^2^ = 0.9964). The citric acid content was then obtained using citric acid as a standard and comparing the standard curves (y = 10,000,000x – 94,537; R^2^ = 1), and the malic acid content was then obtained using malic acid as a standard and comparing the standard curves (y = 1,000,000x + 112,696; R^2^ = 0.9974).

### 4.4. Determination of Lycopene, Carotene and Lutein Content

The content of carotenoid and lycopene was determined according the method of Li et al. [74]. Briefly, tomato fruit was vortexed with methanol and chloroform and centrifugated at 10,000× *g* for 10 min at 4 °C, and the chloroform phase was collected. Then, the residue was repeatedly washed and centrifuged with acetone and chloroform until the residue was white. The supernatant was poured into a separatory funnel, and the lower phase was discarded after standing for stratification. The upper layer was washed with pre-cooled methanol, and the upper phase was discarded after the layers were clarified. Then, the upper phase was collected for testing after repeat washing 2–3 times. The absorbances were measured at 487.5 nm and 502 nm, respectively. The lycopene content was then obtained by using Sudan Red I as a standard and comparing the standard curves (y = 3.1885x; R^2^ = 0.9998). The carotene content was then obtained by using β-carotene as a standard and comparing the standard curves (y = 0.8457x − 0.0208; R^2^ = 0.912). The content of carotenoid and lycopene were calculated according to the following formula: total carotene content = OD_487.5_ × dilution factor × 106/(181 × cuvette thickness × sample weight); lycopene content = (OD_502_ × 181/OD_487.5_ − 42/237) × 100%.

For determining the lutein content, about 1 g tomato fruit was mixed with 30 mL of extraction liquid (the volume ratio of n-hexane: acetone: ethanol: toluene was 10:7:6:7) and 2 mL of 40% KOH methanol solution, and shook for 2 and 1 min, respectively. The above extraction solution was immersed in a water bath at 56 °C for 20 min, and then placed in the dark for 1 h. Subsequently, 30 mL of n-hexane was added to the extraction solution and diluted to 100 mL with 10% sodium sulfate, and placed in the dark for 1 h. Finally, the solution was divided into two layers, and an appropriate amount of supernatant was taken and measured for absorbance at 474 nm with an ultraviolet spectrophotometer (UV-1800, Shimadzu, Kyoto, Japan). The lutein content was then obtained by using Sudan Red I as a standard and comparing the standard curves (y = 0.785x +1.346; R^2^ = 0.9990). The lutein content was calculated according to the following: lutein content (%) = (A × 250)/(W × 236), where A is absorbance and W is sample weight (g).

### 4.5. Determination of Total Phenol, Total Flavonoid and Ascorbic Acid Content

The total phenolic content of the tomato fruit was determined according to the method of Toor et al. [75]. Approximately 1.0 g of the sample was added to 20 mL of ethanol (50%) and mixed well in the test tube, extracted in a water bath for 1 h at room temperature, then filtered and fixed to 25 mL. About 1 mL of the crude extract was mixed with 1 mL 1 mol/L folin-ciocalteau reagent and 1.5 mL of Na_2_CO_3_ solution (7.5%) into a 10 mL test tube, then incubated in the dark at 30 °C for 0.5 h. The absorbance value was measured at 765 nm. The total phenolic content was then obtained by using gallic acid as a standard and comparing the standard curves (y = 11.974x − 0.0199; R^2^ = 0.9965).

The total flavonoid content was measured according to Jia et al. [76]. Pre-chilled 1% HCl-methanol solution was added to 1.0 g of sample and was homogenized on ice, and transferred into a 10 mL tube and centrifuged at 3000× *g* for 20 min after being placed in the dark for 20 min at 4 °C. Then, 0.3 mL NaNO_2_ solution (5%) and 0.3 mL Al(NO_3_)_3_ solution (10%) was added into 1 mL of extraction solution and incubated for 5 and 6 min, respectively. Finally, the absorbance at 510 nm was measured after adding 4 mL 4% NaOH solution and 4.4 mL of distilled water. The total flavonoid content was then obtained by using rutin as a standard and comparing the standard curves (y = 0.5826x − 0.0068; R^2^ = 0.9981).

The ascorbic acid (AsA) content was detected in accordance with the indophenol titration method [41]. Tomato fruit sample was homogenized with 2% oxalic acid and followed by the addition of 30% zinc sulfate and 15% potassium ferrocyanide, and centrifugated at 12,000× *g* for 30 min. The supernatant was collected and mixed with 2,6-dichlorophenol-sodium indophenol solution and xylene. After standing for stratification, the supernatant was used to detect the absorbance at 500 nm. The AsA content was then obtained by using AsA as a standard and comparing the standard curves (y = 0.071x − 0.011; R^2^ = 0.996). The calculation formula is as follows: AsA content (mg/g) = (X value found on the standard curve × 25)/1000.

### 4.6. Determination of the Content of Polyphenols

The content of polyphenols was analyzed according to the method of Castillo et al. with some modifications [77]. Polyphenol compounds were extracted from tomato fruit powder with methanol in an ultrasonic bath for 15 min. The standard samples (Solarbio, Beijing, China) were dissolved separately in methanol solution to prepare a mixed standard. The quantification of this phenolic extract was achieved by external calibration using calibration curves at 5, 2.5, 1.25, 0.625, 0.3125, 0.03125, 0.015625 μg/mL. A HPLC C18 column (250 mm × 4.6 mm, 5 μm, Waters Symmetry) was used under the following condition: flow rate of 1.0 mL/min, column temperature at 30 °C, methanol: acetic acid (100:1, *v*/*v*).

### 4.7. Total RNA Isolation and Gene Expression Analysis

Total RNA was extracted from tomato fruit based on the study of Huang et al. [78]. The cDNA was synthesized by the Evo M-MLV RT-PCR Kit (AG). The qRT-PCR experiments were performed on the Light Cycler^®^ 480 II Real-Time PCR Detection System (Roche, Swiss) using the SYBR Green PCR Master Mix (Takara). The tomato *ACTIN* gene was used as the internal control to calculate the relative expression. The specific primers are listed in Appendix A.

### 4.8. Statistical Analysis

One-way analysis of variance (ANOVA) followed by a Tukey post-hoc test was used to test the differences in parameters among treatments (*p* < 0.05). Each treatment was replicated three times. The log response ratio (LnRR) was calculated as the effect size to evaluate the differences in different treatments as follows: lnRR = ln (Xt/Xc), where Xt was the value of quality parameters in NaHS and HT treatments, and Xc was the value in H_2_O treatment. Since the sample size (*n* = 3) for each quality parameter was consistent, weighted effect sizes were not considered in this study. All statistical analyses and figure plotting were performed on Origin version 2022b (OriginLab Corp., Northampton, MA, USA).

## 5. Conclusions

Our study highlighted the important efficiency of H_2_S in tomato fruit postharvest storage. The evidence provided here suggests that H_2_S application caused an obvious accumulation reduction in soluble sugar (including fructose and glucose) and a sharp increase in the content of soluble protein, organic acid, malic acid and citric acid. Also, exogenous H_2_S up-regulated the expression level of the *PAL5, 4CL, CHS1, CHS2, F3H* and *FLS* genes, thus raising the content of total phenolics and total flavonoids as well as most of phenolic compounds during the storage of the tomato fruit. In addition, H_2_S decreased the content of carotene and lycopene, and down-regulated the expression of the *ZDS, CCD7, CrtL-e-1, psy1, LCY1* and *PDS* genes. The findings provide insight into the significant effects of H_2_S on tomato fruit quality and support the potential application of H_2_S in maintaining tomato fruit nutrients and reduced flavor loss postharvest. However, more molecular mechanisms concerning how H_2_S regulates the quality of tomato fruit need to be explained. For example, the regulatory roles of H_2_S in fruit quality at the post-translational level and the regulatory network of H_2_S still need to be elaborated.

## Figures and Tables

**Figure 1 plants-12-00719-f001:**
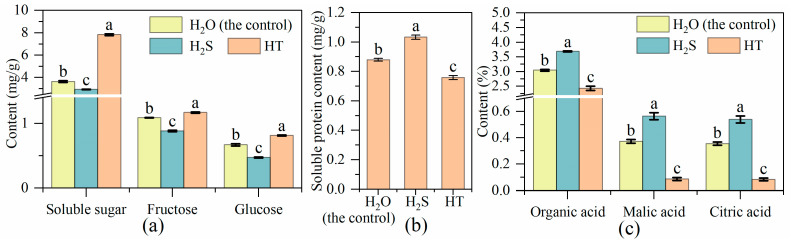
Effects of H_2_S on the content of (**a**) soluble sugar, (**b**) soluble protein and (**c**) organic acids in tomato fruit. Tomato fruit was treated with H_2_O (the control), H_2_S (0.90 mM NaHS solution) and HT (100 mM HT solution). The values are shown as mean ± SE. The different letters indicate the existence of statistical significance among treatment in the same parameter (Tukey post-hoc, *n* = 3, *p* < 0.05).

**Figure 2 plants-12-00719-f002:**
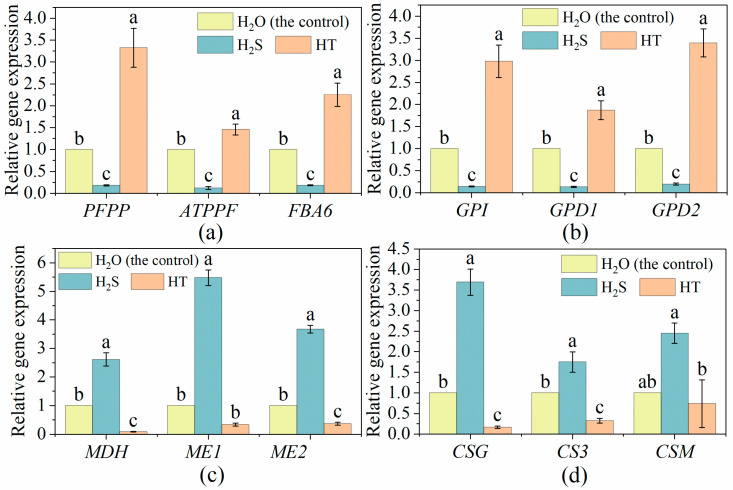
Effects of H_2_S on the expression of (**a**) fructose, (**b**) glucose, (**c**) malic acid and (**d**) citric acid metabolism-related genes in tomato fruit. Tomato fruit was treated with H_2_O (the control), H_2_S (0.90 mM NaHS solution) and HT (100 mM HT solution). The values are shown as mean ± SE. The different letters indicate existence of statistical significance among treatment in the same parameter (Tukey post-hoc, *n* = 3, *p* < 0.05).

**Figure 3 plants-12-00719-f003:**
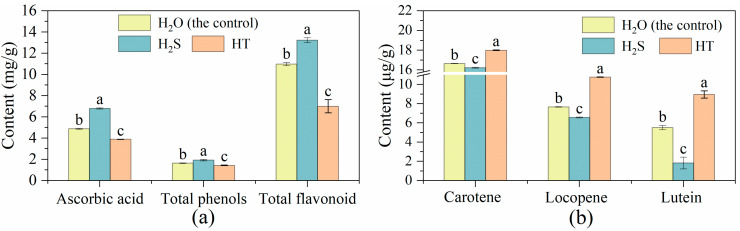
Effects of H_2_S on the content of (**a**) ascorbic acid, total phenols, and total flavonoid and (**b**) carotenes and lycopene in tomato fruit. Tomato fruit was treated with H_2_O (the control), H_2_S (0.90 mM NaHS solution) and HT (100 mM HT solution). The values are shown as mean ± SE. The different letters indicate the existence of statistical significance among treatment in the same parameter (Tukey post-hoc, *n* = 3, *p* < 0.05).

**Figure 4 plants-12-00719-f004:**
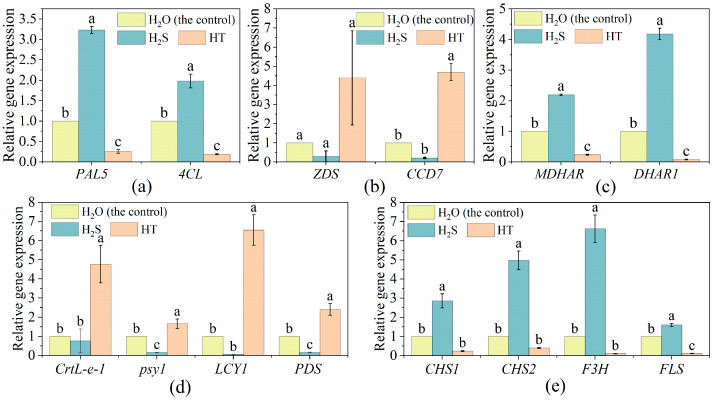
Effects of H_2_S on the expression of (**a**) polyphenols, (**b**) carotenes, (**c**) ascorbic acid, (**d**) lycopene and (**e**) flavonoid metabolism-related genes in tomato fruit. Tomato fruit was treated with H_2_O (the control), H_2_S (0.90 mM NaHS solution) and HT (100 mM HT solution). The values are shown as mean ± SE. The different letters indicate existence of statistical significance among treatment in the same parameter (Tukey post-hoc, *n* = 3, *p* < 0.05).

**Figure 5 plants-12-00719-f005:**
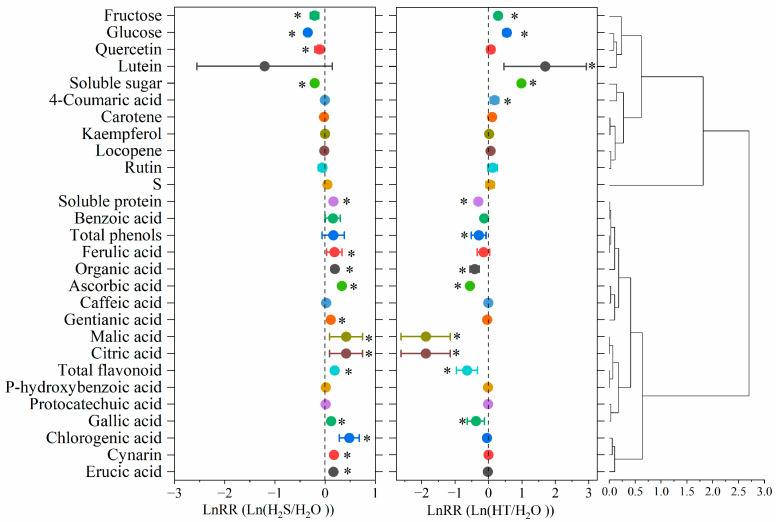
Effect sizes of H_2_S on flavor quality. Error bars indicate 95% confidence intervals (CI). Sample size for each functional trait is consistent (*n* = 3). Effect size greater than zero showed H_2_S and HT have a greater effect on flavor and nutritional quality, and vice versa. Significant effect size at *p* < 0.05 level (*) was shown.

**Table 1 plants-12-00719-t001:** Effects of H_2_S on phenolic compounds in tomato fruit.

Phenolic Compounds (μg/mL)	H_2_O	H_2_S	HT
Rutin	153.656 ± 0.643 ^b^	145.073 ± 2.412 ^b^	173.269 ± 6.199 ^a^
Gentianic acid	34.519 ± 0.182 ^b^	38.484 ± 0.215 ^a^	32.963 ± 0.285 ^c^
Quercetin	16.978 ± 0.17 ^b^	15.142 ± 0.278 ^c^	18.056 ± 0.260 ^a^
Benzoic acid	8.683 ± 0.138 ^b^	10.113 ± 0.198 ^a^	7.594 ± 0.108 ^c^
Gallic acid	8.484 ± 0.018 ^b^	9.540 ± 0.114 ^a^	5.829 ± 0.364 ^c^
Kaempferol	7.887 ± 0.004 ^b^	7.849 ± 0.002 ^c^	8.003 ± 0.012 ^a^
Chlorogenic acid	5.358 ± 0.002 ^b^	8.668 ± 0.400 ^a^	5.09 ± 0.025 ^b^
Ferulic acid	5.332 ± 0.197 ^b^	6.393 ± 0.003 ^a^	4.574 ± 0.042 ^c^
Caffeic acid	5.179 ± 0.0147 ^b^	5.277 ± 0.019 ^a^	5.125 ± 0.019 ^b^
Cinnamic acid	5.144 ± 0.084 ^a^	5.370 ± 0.072 ^a^	5.378 ± 0.075 ^a^
Erucic acid	5.031 ± 0.018 ^b^	5.923 ± 0.011 ^a^	4.900 ± 0.005 ^b^
Cynarin	4.993 ± 0.025 ^b^	5.937 ± 0.095 ^a^	4.960 ± 0.034 ^b^
Protocatechuic acid	4.605 ± 0.004 ^b^	4.636 ± 0.004 ^a^	4.522 ± 0.004 ^c^
P-hydroxybenzoic acid	4.552 ± 0.012 ^a^	4.605 ± 0.031 ^a^	4.458 ± 0.027 ^b^
4-Coumaric acid	4.302 ± 0.001 ^b^	4.270 ± 0.014 ^b^	5.141 ± 0.148 ^a^

The values are shown as mean ± SE. The different letters indicate existence of statistical significance among treatment in the same parameter (Tukey post-hoc, *n* = 3, *p* < 0.05).

## Data Availability

The data presented in this study are available on request from the corresponding author.

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
