# Peer review of "Effects of Hydrogen Sulfide on Sugar, Organic Acid, Carotenoid, and Polyphenol Level in Tomato Fruit"

_plants, 2023, doi:10.3390/plants12040719_

Round 1

Reviewer 1 Report

The MS by Zhang et al. presents interesting findings. However, the MS has significant errors in English language and grammar. Therefore, I would recommend authors to revise the MS thoroughly and correct the grammar errors seriously. Suggestions are appended below:

I abstract L17 authors should replace the word ‘enhancing’ its better to use ‘confirming’ or ‘suggesting’.

Introduction should be improved. Most of the introduction is written on H2S. The global data on postharvest yield losses of tomato should be added. Also add the reason of flavor loss in tomato. L67/68 the preharvest and postharvest factors should be included and explain in detail.

Sub heading 3.2 is not meaningful. It should be changed to “Effects of H2S on the expression of genes related with soluble sugar and organic acids metabolism in tomato fruit”. Similar changes should be made in 3.4 as: Effects of H2S on the expression of genes related with polyphenols, flavonoid, ascorbic acid and carotene metabolism in tomato fruit.

-          There are several syntax errors e.g. L 182 ‘substances’ ‘molecules’ should be ‘substance’ and ‘molecule’.

-          L188-190: During cultivated tomato fruit maturity stage, fructose and glucose as the main sugars were increased with the consume of sucrose. In this sentence ‘consume’ should be changed to ‘consumption’.

-          The sentence is technically incorrect. “Three replicates were repeated independently in each experiment”. It should be: “Each treatment was replicated three times” or “Three replicates were considered independently for each treatment”.

-          I would recommend authors to carefully check the English grammar of the whole MS.

-          Discussion should be improved by providing a mechanistic approach through interconnecting the studied parameters.

Author Response

Please see the attachment for the modification.

Reviewer 2 Report

Comments to the authors,

The effect of hydrogen sulfide on fruit quality is very interesting and has a lot of benefits. The manuscript discuss some effects of the molecule on tomato fruit components, however, it is not accurate to state that effects the fruit quality according to the discussed data.

There are some comments that should be considered:

1. The manuscript title is incorrect and mislead the reader. According to the title the readers expect information of sensorial evaluation of the fruit flavor (a taste panel, e-nose, e-tongue). Therefore, the manuscript title must be change and describe the information presented (chemical molecules, genes related to sugar metabolism etc.).

Line 33: None of the citations related to postharvest preservation (4,8,9). Similar mistake happened in more citation (the citation is incorrect). Please look again and fix similar mistakes.

Line 89: it is advised to rephrase the sentence. Do not begin it with: Fig. 1C. Provided…

Figure 1 a,c, and all the other similar figures- it will be easier to understand the effect of the treatment if you change the figures that the soluble sugar, fructose and glucose (figure 1a) will be at the X- axis, while the treatments will be the legends. Please change that in accordance.

Lines 101, 103, 104, etc- write the full name of the gene or any abbreviation at its first appearance in the text.

Lines 101, 103, 104- add information and explain why you chose the specific genes.

Figure 2a: are you sure that the expression of H2S PFPP gene is not significantly different than the H2O gene expression?

Line 134: delete the word obviously.

Line 134: It is preferred to rephrase the sentence: the expression of the genes was significantly higher in HT treated fruit (I am not sure that is correct to say that HT treatment increased the level….).

Line 142: Effects of H2S on the expression level of polyphenols, flavonoid, ascorbic acid and carotene metabolism-related genes in tomato fruit

Line 152: you should rephrase and just describe the results. Do not write: HT treatment declined their content. Instead it should be better to write: The content of the… was lower at the HT treated tomatoes.

Table 1: it will be better to re-organize the order of the compounds according to their concentration or any logical order.

Line 159- please explain better the way of calculation in figure 5. It is simply the ratio between compounds. Correct?

Line 159-166- you should add information related to the branches if the phylogenetic tree.

Line 177: why did you choose citation 49? Is this the best for that? Consider to replace it.

Line 178: fruit not fruits

Line 183: affects instead of regulates. It is not correct to say that it regulates sweetness.

Line 191-192: please rephrase. What do you mean: in keeping with the …

Line 197-198: Zhong et al. [61] and Li et al. [18] also showed that H2S raised citric acid content in tomato and kiwifruit, respectively.

Line 210-213: what is the point that you want to say? There is not enough discussion. No specific outcome…

Line 227-239: most of this part describes the results. The discussion part should not repeat the results part so extensively. Similar repeat that describe the results part could be find in line 244-246 and other.

I could not find discussion related to the soluble protein content. Please add.

Materials and methods:

Line 258: add ripening indices of the tomatoes: brix, color, weight, more…

Line 261-267: please describe and explain better how you produced H2S gas. How did you measure its concentration? What was it?

Line 270-273: the samples were collected from 0-5 days after storage but the results describe only one point. There are no measurements describe the change of compounds with the time. What was the specific sampling stage? This must be clear!

Line 274: was biological replicate a signal fruit? It is not clear.

Line 317-318: did you verify with a standard of carotene? Lycopene? Any standard curve?

Line 334: folin? (not florin)

Line 335: incubated instead of reacted

Line 341: what do you mean- successively?

Line 343-344: what was served as standard?

Line 345-350: any standard curve?

General comment: Add the information regard standard curve and compound for each of the measurements you tested.

Important general comment: it is possible that the H2S treatment delayed the ripening of the tomato. Therefore you must add information related to the ripening stage of the tomato when you compare the measurements. This should include: Brix (%), total acid (%), color using colormeter (Minolta), stiffness, photo of the tomatoes when sampled.

Author Response

(The authors gave the same response as above.)

Reviewer 3 Report

The article describes the effect of hydrogen sulfide on the taste of tomato fruits. The authors evaluated the effectiveness of H2S in the taste qualities of tomato fruits in the post-harvest period.

The article is well structured, the results are presented clearly.

There is a small note: In conclusion, indicate directions for further research. The "materials and methods" section should be placed before the "results" section.

Author Response

(The authors gave the same response as above.)

Round 2

Reviewer 1 Report

The authors have revised the MS as per the suggestions. 

Reviewer 2 Report

Dear Authors,

Thank you for your careful improvement of the manuscript.

You significantly improved the manuscript and it is much brighter.

Please, read it again carefully to find minor mistakes as you may see in the legends of figure 1 and 3 it was written cortrol (instead of control).

Good Luck!
